# Regional trends, spatial patterns and determinants of health facility delivery among women of reproductive age in Nigeria: A national population based cross-sectional study

Tope Olubodun[1]*, Olorunfemi Akinbode Ogundele[2], Turnwait Otu Michael[3], Oluyemi Adewole Okunlola[4], Ayodeji Bamidele Olubodun[5], Semiu Adebayo Rahman[6]

1 Department of Community Medicine and Primary Care, Federal Medical Center, Abeokuta, Ogun State, Nigeria, 2 Department of Community Medicine, University of Medical Sciences, Ondo City, Nigeria, 3 Department of Sociology, University of Johannesburg, Johannesburg, South Africa, 4 Department of Mathematical and Computer Sciences, University of Medical Sciences, Ondo City, Nigeria, 5 Olabisi Onabanjo University Teaching Hospital, Sagamu, Nigeria, 6 Department of Demography and Social Statistics, Obafemi Awolowo University, Ile Ife, Osun State, Nigeria

* olubeduntope@gmail.com

## Abstract

### Background

Globally, about 810 women die daily from pregnancy and childbirth complications, and the burden is highest in Africa. The United Nations sustainable development goal has a maternal mortality ratio (MMR) target of 70 per 100,000 live births by 2030. Nigeria, the largest country in Africa, has an MMR of 512 per 100,000 live births, thus there is need for intensified efforts to reduce maternal deaths in the country. Proper utilisation of maternal health services including health facilities for delivery is crucial to achieving this. This study assesses the regional trends, spatial patterns and determinants of health facility delivery among women of reproductive age in Nigeria.

### Methods

This is a weighted secondary analysis of the Nigerian Demographic and Health Survey (NDHS). The sample included women who had a live birth in the preceding 5 years of the NDHS 2008, 2013 and 2018. Bivariate analysis and multilevel logistic regression were carried out to assess the determinants of health facility delivery. Trends analysis was done using bar graphs and spatial analysis showed the distribution of health facility delivery by State.

### Results

Forty-one percent of women delivered their last child in a health facility. The proportion of women who delivered at a health facility increased from 2008 to 2018 for all regions, with

**Data Availability Statement:** The data underlying the results presented in the study are available after registration and request from measure DHS website (https://dhsprogram.com).

**Funding:** The author(s) received no specific funding for this work.

**Competing interests:** The authors have declared that no competing interests exist.

exception of the South-south region. Determinants of facility-based delivery include; ethnicity, level of education, wealth index, exposure to mass media (AOR 1.34; 95% CI 1.20–1.50), number of childbirths, number of antenatal visits (AOR 4.03; 95% CI 3.51–4.62), getting a companion to go the health facility (AOR 0.84; 95% CI 0.72–0.98), community level poverty, community level of female education, community distance to health facility, and geographical region.

## Conclusion

There is an urgent need to deploy appropriate strategies and programme to improve health facility delivery in Nigeria.

## Introduction

Delivery in a health facility by skilled birth attendants and with available life-saving commodities and facilities reduces maternal morbidity and morbidity significantly [1, 2]. Health facility delivery also reduces stillbirth rate and neonatal morbidity [3, 4]. Recognising the importance of health facility delivery, the World Health Organization (2023) recommends that all births take place at health facilities with the assistance of skilled birth attendants. In 2021, 84% of births worldwide happened in health facilities [5]. The proportion of facility delivery varies greatly among countries. Almost all newborns (99%) are born at a health facility in developed countries [5]. Only 64% of babies in Sub-Saharan Africa (SSA) have skilled birth assistance during delivery [5]. Others give birth at home with the help of inexperienced birth attendants, family members, or self-delivery [6]. In Sierra Leone for example only 45.2% utilise skilled birth assistance during delivery, in Mali 39.9%, and in Niger 32.6% [7].

In Nigeria, only 41% of Nigerian women deliver in health facilities [8]. Variations in health facility delivery exist across Nigerian geopolitical zones, ranging from 16% in the Northwest zones to 81% in the Southeast zones [9]. Tackling the challenges associated with health-care delivery is critical, especially in Nigeria, where the crude birth rate is an estimated 38 births per 1000 women [9], with a total fertility rate of 5.3 [9]. Disregarding adequate actions to improve health facility delivery in Nigeria will exacerbate the country's already high maternal mortality rate (MMR) of 512 per 100,000 live births, which makes Nigeria still a long way from meeting the Sustainable Development Goal of fewer than 70 maternal deaths per 100,000 live births by 2030 [10].

Sociocultural, economic, and policy changes can affect maternal healthcare utilization over time. Educational attainment of women has improved over time in Nigeria, even though educational attainment is still low. A study reported that women of reproductive age with secondary/higher education was 31.8% in 2008 [11], and another study reported a prevalence of 38.7% in 2018 [12], and both studies used secondary analysis from the NDHS. Sociocultural factors that can influence health seeking behaviour and maternal healthcare utilization include religious and cultural beliefs [13, 14]. It is not clear how this has evolved over time as there is paucity of research that has studied such trends, moreso, it may be difficult to assess using quantitative approaches. Regarding the economy, Nigeria has experienced some economic down turns. Annual GDP per capita growth rate in 2008 was 3.9%, 3.8% in 2013, and -0.6% in 2018 [15]. This could affect people's ability to pay for healthcare.

A myriad of programmes have been instituted in Nigeria, over time to improve maternal health. To enhance maternal health in Nigeria, the Midwives Service Scheme and the Subsidy

Reinvestment and Empowerment Programme (SURE-P) were introduced [16, 17]. The Midwives service scheme was introduced in 2009 to improve availability of skilled birth attendants in rural areas of the country [16]. The program engages retired midwives, unemployed and newly graduated midwives to work temporarily in rural areas. In 2012, SURE-P was introduced in a bid to re-invest fuel subsidy funds into social safety net programs including improving maternal health. SURE—P includes a conditional cash transfer for mothers attending at least four antenatal visits, delivering in a health facility, and also attending postnatal care visits, health facility renovations and staffing, ensuring supply of essential maternal health commodities, and community mobilization through village health workers and community leaders [17]. Both programmes had some successes [16, 17], however, in 2018, less than half of women in Nigeria delivered at a health facility [8]

Some studies have been carried out using nationally representative data to assess health facility delivery in Nigeria. Our study adds to the body of knowledge from these studies. A study carried out in 2013, identified the determinants of health facility delivery among Nigerian women using data from the 2008 NDHS [11]. Another study identified factors associated with home delivery among Nigerian women and conducted a spatial analysis to capture the locations where home delivery is prevalent in the country using data from the 2013 NDHS [18]. A study assessed the determinants of antenatal care, health facility delivery and postnatal care among women in Nigeria using the 2018 NDHS, but only included individual level variables in analysis and did not account for the hierarchical nature of the NDHS [12]. Yet, another study assessed the determinants of health facility delivery with emphasis on community level factors but only examined a few individual factors [19].

Our study is different from these studies as we use the most recent DHS to provide more recent estimates we examine the regional trends, determinants, and spatial patterns of health facility delivery rather than home delivery, we use a multilevel approach which takes into account the hierarchical nature of the DHS, and we include a significant number of individual level factors and community level factors. Our study thus provides recent estimates of the determinants of health facility delivery, examines the trends in health facility delivery over ten years and across the six geopolitical zones, and uses spatial analysis to demonstrate parts of the country with high, medium, and low prevalence of health facility delivery. These findings will provide evidence that can guide policy and programming in maternal health in Nigeria.

## Methods

### Data source

Data from Nigeria Demographic and health surveys were used in this study. The trend analysis made use of data from NDHS 2008, NDHS 2013 and NDHS 2018 to analyze the trend of health facility delivery across these years, and across the six geo-political zones. Analysis of the determinants of health facility delivery and the spatial analysis of the distribution of health facility delivery across States was done using data from the most recent NDHS i.e., NDHS 2018. Demographic and Health Surveys (DHS) are nationally representative household surveys that provide data for a wide range of indicators in the areas of population, health, and nutrition [20]. They are usually carried out every five years and the data can be used for monitoring and impact evaluation and research [20].

NDHS uses a two-stage cluster sampling approach to select respondents from rural and urban areas in Nigeria and from the 36 States and the FCT. The primary sampling units (PSU)/clusters are the enumeration areas (EAs) from the 2006 census and the Population and Housing Census of the Federal Republic of Nigeria (NPHC), conducted in 2006 provides the sampling frame [9].

## Study variables

**Outcome variable.**   The outcome variable is facility-based delivery of the most recent birth. Births in a public or private health facility was defined as 'utilized health facility delivery' and coded as 1, while those who delivered at home or elsewhere were defined as 'not utilizing health facility delivery and was coded as 0.

**Exposure variables.**   Age at last childbirth was derived by subtracting the date of birth of mother (in century month code CMC) from date of birth of child (in century month code) and dividing by 12. Age group was then categorized as 15–19 years, 20–29 years, 30–39 years, 40–49 years. Women that were never in union and those that were formerly in union/living with a man were categorized as 'not married' and those currently in union/living with a man were grouped as 'married'. Religion was categorised as Christianity, Islam, and Traditional religion. Ethnicity was categorised as Hausa/Fulani, Yoruba, Ibo, and Others. Level of education was expressed as no education, primary, secondary, and higher. Respondents' employment status was categorized as 'currently working' and 'not currently working'. Wealth index was generated as a tertile of the wealth index factor score into poor, middle and rich.

Mass media exposure was generated from exposure to television, radio, and newspaper. Mass media exposure was defined as frequent exposure for those with at least once a week exposure to television, radio or newspaper, and No exposure/infrequent exposure for those who had no access to any of these or less than once a week exposure to any of these. Wanted index pregnancy was recoded as 'wanted' or 'not wanted'. Number of childbirths was categorized as 1–2, 3–4, $\geq$ 5. Number of antenatal care (ANC) visits was categorized as less than four ANC visits, and at least four ANC visits. Companionship to health facility was categorized as being 'a big problem' and 'not a big problem'. Woman's participation in healthcare decision was recoded as participate and does not participate. Partner's education was expressed as no education, primary, secondary, and higher.

The following factors were considered at community level: community level poverty, community level women's education, community distance to health facility, place of residence (urban or rural) and region. Region was used as provided in the NDHS dataset as Northcentral, Northeast, Northwest Southeast, South-south and Southwest. Other community level variables were computed by aggregating individual characteristics at the cluster level, dividing the measure into tertiles, and categorizing as low, medium and high. Similar procedure has been widely applied to derive community variables in DHS datasets [12, 18]. Community level poverty was defined as the proportion of women who are from the poorest communities. Community women's education was defined as proportion of women from community with at least secondary education. Community distance to health facility was defined as the proportion of women for whom distance to health facility is a big problem, aggregated at cluster level.

The variables—religion, respondent's education status, respondent's employment status, companionship to health facility, partner's education, place of residence and region—were used for analysis as they were originally in the NDHS. All other variables were recoded from existing variables.

## Data analysis

Data analysis was done using Stata (17, StataCorp LLC, College Station, TX, USA). In DHS analysis, in order to adjust for multi-level cluster sampling design and non-response, individual women's survey weights are needed. Therefore, we adjusted for sampling weights, clustering, and stratification. Descriptive analysis included the trends analysis and frequency distribution to present background characteristics. Trend analysis was presented in bar charts.

Bivariate analysis was done using Chi-Square test to test the association between the independent variables and place of delivery.

**Multilevel analysis.** Taking into consideration the hierarchical structure of the DHS, a mixed effect multilevel logistic regression analysis, a Generalized Linear Mixed Models (GLMM) was done to identify the determinants of health facility delivery. Standard regression assumes independence of observations (lack of correlation). The DHS however uses a two-stage cluster sampling procedure and observations in the DHS data are not truly independent, due to the effect of clustering. Individuals from a cluster are likely to have similar characteristics and may be different from individuals from other clusters. In the NDHS, individuals are nested within households and households are nested within communities(clusters), which are nested within states. Running a standard logistic regression analysis, may lead to incorrect estimates. Therefore, we did a multilevel analysis to account for the effect of clustering/the hierarchical nature of the NDHS data, to ensure more accurate estimates [21].

All independent variables were statistically significant with p values ≤ 0.05, so all independent variables were included in multivariate analysis. Observations with missing data were excluded from the multivariate analysis. Variance inflation factor was computed prior to the multilevel regression to test for multicollinearity, and a value 3.80 was gotten which is less than 5. The variable marital status was omitted due to collinearity and was not included in the multivariate analysis.

**Spatial analysis.** To create spatial maps for health facility delivery coverage in Nigeria, a sampling dataset was used and analysed using the QGIS 3.321 (https://qgis.org/en/site/). In order to normalize the dataset and make it easier to integrate into the database and visualize in QGIS, the data structure was created using Google Sheets. The attribute data was then combined with the spatial data using the Join Attribute by Location Tool in QGIS. This produced a database that contained the health facility delivery results along with the Nigeria shape file sourced from the humanitarian data exchange (HDX) (https://data.humdata.org/dataset/cod-ab-nga?). The HDX is an open data platform managed by the United Nations Office for the Coordination of Humanitarian Affairs (OCHA) through its Centre for Humanitarian Data and dataset obtained therein are free for use. To graphically represent each point of the attribute data for the variables in QGIS, the Equal Count (Quantile) mode and five classes were employed. The classes show the proportion of each attribute's data that is contained in the database, and this information was displayed using different colour ramps. This step is crucial in the data cleaning process and helps to ensure that the data is accurately represented in the visualization.

### Ethical approval

Being a secondary data analysis, ethical approval was not required for this study. We registered and obtained permission to download the datasets from the measure DHS website. However, in the primary studies—The NDHS 2008, 2013, and 2018, the survey protocols were approved after review by the ICF Institutional Review Board and the National Health Research Ethics Committee of Nigeria (NHREC). Informed consent was obtained, and all methods were performed in accordance with the Declaration of Helsinki.

## Results

Forty-one percent of women delivered their last child in a health facility. Most (49.26%) of the respondents were of the 20–29 years age group. Sixty-one percent were Muslims, 46.42% were of the Hausa/Fulani tribe, 44.44% had no formal education and 68.39% were employed. Majority (60.12%) of the respondents were not exposed to mass media, or had infrequent exposure,

majority (87.85%) desired the index pregnancy, and more than half (57.79%) had at least four antenatal visits. Majority (83.60%) found companionship to the health facility as "not a big problem" but only 38.40% of the women participated in decisions regarding their health. Most of the women (60.24%) resided in rural areas (Table 1).

The proportion of women who delivered at a health facility increased from 2008 to 2018 for all regions, with exception of the South-south region where facility-based delivery remained almost constant in 2013 and 2018. The Northwest had the lowest prevalence of health facility delivery of 9.34% in 2008, 12.8% in 2013 and 16.36% in 2018, while the Southeast had the highest prevalence of 73.84% in 2008, 78.83% in 2013 and 80.8% in 2018. In Nigeria, health facility delivery increased slightly from 36.57% in 2008 to 37.44% in 2013 and then to 41.14% in 2018 (Fig 1).

Table 2 shows that all the independent variables showed statistically significant association with the outcome variable 'health facility delivery'. Higher proportion of women of the age group 20–29 years (41.37%) and 30–39 years (46.07%) delivered in a health facility compared to older women aged 40–49 years (36.03%) and younger women aged 15–19 years (29.44%). A higher proportion of Christian women (65.75%) than Muslim women (25.98%) and women of traditional religion (29.14%) delivered in a health facility. Igbo women had the highest utilization of health facility for delivery (81.30%), followed by Yoruba women (75.29%), then other minority tribes (46.81%) and Hausa/Fulani women (17.46%). While majority of women with higher than secondary education (87.71%) delivered in a health facility, only 15.14% of women without formal education delivered in a health facility. While 69.69% of women from the rich wealth tertile delivered in a health facility, only 13.99% of women from the poor wealth tertile delivered in a health facility (Table 2).

A greater proportion of women with frequent exposure to mass media (60.10%) delivered in a health facility compared to those with infrequent or no exposure to mass media (28.58%). The proportion of health facility delivery was higher among women who did not desire the index pregnancy (52.21%) than for women who desired the index pregnancy (39.62%). The proportion that reported delivery of their last child in a health facility was higher among women with at least four antenatal visits (59.31%) than for women with less than four visits (14.97%). A greater proportion of women who reported companionship to health facility as not a big problem and those who participate in decisions regarding their health delivered in a health facility than women who reported companionship to health facility as a big problem and those who did not participate in decisions regarding their health, respectively.

The proportion of women who had their most recent birth in a health facility was highest for women whose partner had higher than secondary school education (71.13%) and lowest for women whose partners had no formal education. Communities with a low proportion of poor people, a low proportion of uneducated people, and a low proportion of people who considered distance to health facility as a big problem had higher rates of health facility delivery. A higher proportion of women in urban (62.12%) compared with women living in rural areas (27.30%) delivered in a health facility. The Northwest region had the lowest proportion (16.36%) of women delivering in health facilities while the Southeast region (80.80%) had the highest proportion of women delivering in health facilities (Table 2).

In Model 4, the final model which consisted of individual and community variables, Ibo women had 3 times higher odds of delivering in a health facility than Hausa/Fulani women (AOR 3.08; 95% CI 2.11–4.49) and women from ethnic minorities had 50% higher odds of delivering in a health facility compared with Hausa/Fulani women (AOR 1.50; 95% CI 1.24–1.83). As level of education increased, the odds of delivering in a health facility also increased and similarly as partners level of education increased, the odds of delivering in a health facility increased. Compared with women from the poor wealth index, women of the middle wealth

**Table 1. Sample characteristics and prevalence of health facility delivery among women of reproductive age in Nigeria.** (NDHS 2018) (N = 21,792).

| Variables | Frequency | Percentage |
|---|---|---|
| **Facility based delivery** | | |
| Utilized health facility delivery | 9001 | 41.14 |
| Did not utilize health facility delivery | 12791 | 58.86 |
| **Age at last childbirth** | | |
| 15–19 | 2631 | 12.20 |
| 20–29 | 10685 | 49.26 |
| 30–39 | 7180 | 32.76 |
| 40–49 | 1296 | 5.781 |
| **Marital status** | | |
| Not married | 1373 | 5.81 |
| Married | 20419 | 94.19 |
| **Religion** | | |
| Christianity | 8929 | 38.08 |
| Islam | 12687 | 61.39 |
| Traditional religion/others | 176 | 0.53 |
| **Ethnicity[m]** | | |
| Hausa/Fulani | 9226 | 46.42 |
| Yoruba | 2357 | 12.61 |
| Ibo | 2836 | 12.62 |
| Others | 7355 | 28.35 |
| **Level of education** | | |
| No formal education | 9527 | 44.44 |
| Primary education | 3410 | 15.03 |
| Secondary education | 7064 | 31.77 |
| Higher | 1791 | 8.76 |
| **Employment status** | | |
| Unemployed | 6977 | 31.61 |
| Employed | 14815 | 68.39 |
| **Wealth index** | | |
| Poor | 7264 | 31.72 |
| Middle | 7264 | 32.29 |
| Rich | 7264 | 35.99 |
| **Exposure to mass media** | | |
| No exposure/infrequent exposure | 13,446 | 60.12 |
| Frequent exposure | 8346 | 39.88 |
| **Wanted index pregnancy** | | |
| Wanted | 19054 | 87.85 |
| Not wanted | 2738 | 12.15 |
| **Number of childbirths** | | |
| 1–2 | 7493 | 35.06 |
| 3–4 | 6233 | 28.16 |
| ≥ 5 | 8066 | 36.79 |
| **Number of antenatal visits[m]** | | |
| < 4 visits | 9158 | 42.21 |
| At least 4 visits | 12307 | 57.79 |
| **Companionship to health facility** | | |

(*Continued*)

**Table 1.** (Continued)

| Variables | Frequency | Percentage |
|---|---|---|
| A big problem | 3574 | 16.40 |
| Not a big problem | 18218 | 83.60 |
| **Participates in healthcare decision**. | | |
| Participates | 8309 | 38.40 |
| Does not participate | 13483 | 61.60 |
| **Partner's highest level of education**[m] | | |
| No formal education | 7141 | 36.14 |
| Primary education | 2897 | 13.95 |
| Secondary education | 7060 | 34.46 |
| Higher | 3039 | 15.44 |
| **Community poverty** | | |
| Low | 10292 | 48.89 |
| Medium | 4299 | 20.71 |
| High | 7201 | 30.41 |
| **Community women's education** | | |
| Low | 7285 | 34.29 |
| Medium | 7253 | 32.73 |
| High | 7254 | 32.99 |
| **Community distance to health facility** | | |
| Low | 7281 | 35.28 |
| Medium | 7319 | 33.54 |
| High | 7192 | 31.19 |
| **Place of residence** | | |
| Urban | 7710 | 39.76 |
| Rural | 14082 | 60.24 |
| **Region** | | |
| Northcentral | 3875 | 13.83 |
| Northeast | 4506 | 17.63 |
| Northwest | 6309 | 34.89 |
| Southeast | 2365 | 9.75 |
| South-south | 2174 | 9.21 |
| Southwest | 2563 | 14.69 |

[m]Variables with missing data

index had 32% higher odds of delivering in a health facility (AOR 1.32; 95% CI 1.12–1.54) and women of the rich wealth index had 90% higher odds of delivering in a health facility (AOR 1.90; 95% CI 1.55–2.33). Women with frequent exposure to mass media had 34% higher odds of delivering in a health facility (AOR 1.34; 95% CI 1.20–1.50). The odds of delivering in a health facility reduced as number of childbirths increased. Women with at least four antenatal visits were 4 times more likely to deliver in a health facility than women with less than four antenatal visits (AOR 4.03; 95% CI 3.51–4.62). Women who reported getting a companion to go the health facility was a big problem were less likely to deliver in a health facility (AOR 0.84; 95% CI 0.72–0.98) (Table 3).

Women from communities with medium level of community poverty (AOR 0.74; 95% CI 0.59–0.93). and those from communities with high level off community poverty (AOR 0.63;

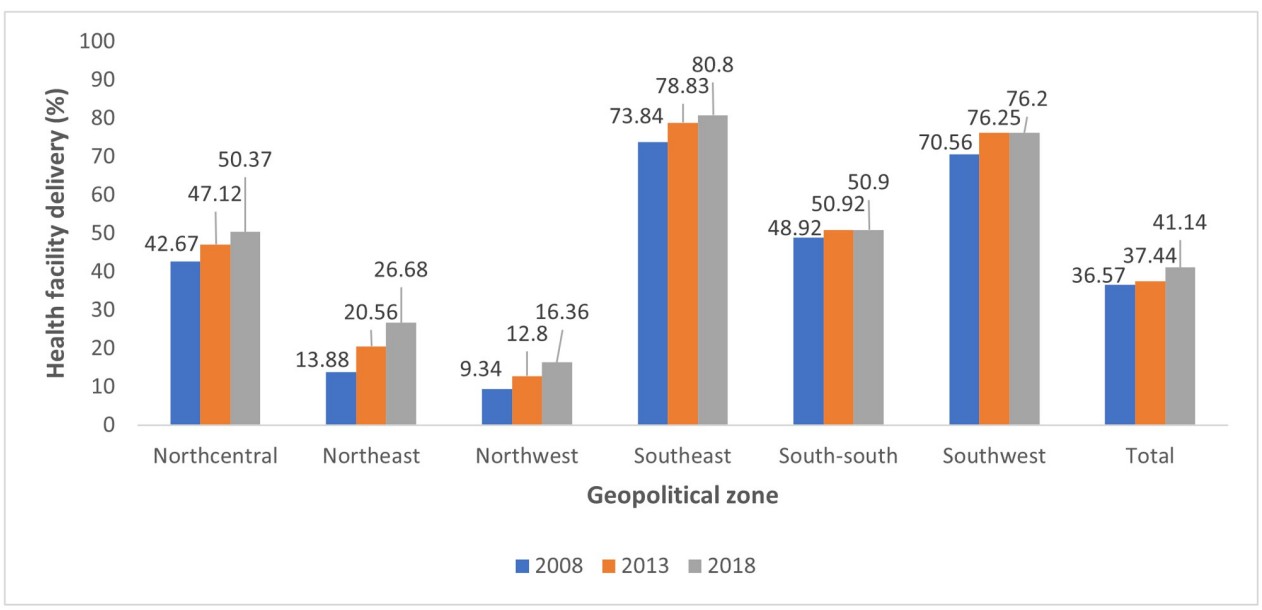

**Fig 1. Regional trends of health facility delivery in Nigeria (NDHS 2008, NDHS 2013, NDHS 2018).**

95% CI 0.48–0.83) had lesser odds of delivering in a health facility, compared to women from communities with low level of community poverty. Compared with women from communities with low level of female education, women from communities with medium level of female education (AOR 1.72; 95% CI 1.32–2.24). and those from communities with high level of female education (AOR 2.65; 95% CI 1.88–3.72) had higher odds of delivering in a health facility. Women from communities where a high proportion of women considered distance to health facility as a big problem were less likely to use health facility for child delivery (AOR 0.71; 95% CI 0.56–0.88). Women from the Northeast, Northwest and South-south regions were less likely to use health facility for child delivery, compared with women from the North-central region (Table 3).

Age at last childbirth, religion, employment status, wanted index pregnancy, participation in healthcare decision and place of residence did not show statistically significant relationship with place of delivery (Table 3).

Fig 2 shows the prevalence of health facility delivery among women of reproductive age in Nigeria (Fig 2).

## Discussion

This study assessed the regional trends, determinants, and spatial patterns of health facility delivery in Nigeria. Our study reveals that only four in ten of women deliver in a health facility in the country. This finding is significant as it demonstrates that poor facility delivery among women of reproductive age exists in the country and is in keeping with the findings of Bolarinwa et al., who similarly used the NDHS to assess health facility delivery among women of reproductive age [8]. In addition, this finding is equally essential because the place of delivery impacts the quality of maternal and child health services received and may thus emphasize the need for improving existing strategies and establishment of new programmes to improve the uptake of facility delivery among reproductive-age women.

**Table 2. Bivariate analysis of factors associated with health facility delivery among women of reproductive age in Nigeria (NDHS 2018) (N = 21,792).**

| Variables | Facility based delivery | | p-value | Crude odds Ratio (95% CI) |
|---|---|---|---|---|
| | Did not utilise health facility delivery | Utilised health facility delivery | | |
| | Freq (%) | Freq (%) | | |
| **Age at last childbirth** | | | | |
| 15–19 years | 1821 (70.56) | 810 (29.44) | <0.0001 | ref |
| 20–29 years | 6289 (58.63) | 4396 (41.37) | | 1.69(1.51–1.90)** |
| 30–39 years | 3876 (53.93) | 3304 (46.07) | | 2.05(1.81–2.32)** |
| 40–49 years | 805 (63.97) | 491 (36.03) | | 1.35(1.13–1.61)* |
| **Marital Status** | | | | |
| Not married | 687 (48.09) | 686 (51.91) | <0.0001 | ref |
| Married | 12104 (59.52) | 8315 (40.48) | | 0.63(0.54–0.73)** |
| **Religion** | | | | |
| Christianity | 3306 (34.25) | 5623 (65.75) | <0.0001 | ref |
| Islam | 9357 (74.02) | 3330 (25.98) | | 0.18(0.16–0.21)** |
| Traditional | 128 (70.86) | 48 (29.14) | | 0.21(0.12–0.39)** |
| **Ethnicity**[m] | | | | |
| Hausa/Fulani | 7589 (82.54) | 1637 (17.46) | <0.0001 | ref |
| Yoruba | 550 (24.71) | 1807 (75.29) | | 14.40(11.76–17.64)** |
| Ibo | 559 (18.7) | 2277 (81.3) | | 20.55(16.56–25.51)** |
| Others | 4087 (53.87) | 3268 (46.81) | | 4.16(3.59–4.82)** |
| **Level of education** | | | | |
| No formal education | 8065 (84.86) | 1462 (15.14) | <0.0001 | Ref |
| Primary education | 1960 (58.47) | 1450 (41.53) | | 3.98(3.49–4.55)** |
| Secondary education | 2537 (35.49) | 4527 (64.51) | | 10.19(8.95–11.60)** |
| Higher | 229 (12.29) | 1562 (87.71) | | 39.994** |
| **Employment status** | | | | |
| Unemployed | 4878 (70.89) | 2099 (29.11) | <0.0001 | Ref |
| Employed | 7913 (53.29) | 6902 (46.71) | | 2.13(1.95–2.34)** |
| **Wealth index** | | | | |
| Poor | 6171 (86.01) | 1093 (13.99) | <0.0001 | Ref |
| Middle | 4446 (64.00) | 2818 (36.00) | | 3.46(2.98–4.02)** |
| Rich | 2174 (30.31) | 5090 (69.69) | | 14.14(12.02–16.61)** |
| **Exposure to mass media** | | | | |
| No exposure/infrequent exposure | 9457 (71.42) | 3989 (28.58) | <0.0001 | Ref |
| Frequent exposure | 3334 (39.90) | 5012 (60.10) | | 3.76(3.41–4.16)** |
| **Wanted index pregnancy** | | | | |
| Wanted | 11459 (60.38) | 7595 (39.62) | <0.0001 | Ref |
| Not wanted | 1332 (47.79) | 1406 (52.21) | | 1.66(1.49–1.86)** |
| **Number of childbirths** | | | | |
| 1–2 | 3777 (49.59) | 3716 (50.41) | <0.0001 | Ref |
| 3–4 | 3448 (55.34) | 2785 (44.66) | | 0.79(0.73–0.87)** |
| ≥ 5 | 5566 (70.38) | 2500 (29.62) | | (0.38–0.45)** |
| **Number of antenatal visits**[m] | | | | |
| < 4 visits | 7742 (85.03) | 1416 (14.97) | <0.0001 | Ref |
| At least 4 visits | 4941 (40.69) | 7366 (59.31) | | 8.28(7.45–9.21)** |
| **Companionship to health facility** | | | | |
| A big problem | 2600 (70.31) | 974 (29.69) | <0.0001 | Ref |
| Not a big problem | 10191 (56.72) | 8027 (43.28) | | 1.81(1.57–2.08)** |

*(Continued)*

**Table 2.** (Continued)

| Variables | Facility based delivery | | p-value | Crude odds Ratio (95% CI) |
| --- | --- | --- | --- | --- |
| | Did not utilise health facility delivery | Utilised health facility delivery | | |
| | Freq (%) | Freq (%) | | |
| **Participates in healthcare decision.** | | | | |
| Participates | 3585 (42.13) | 4724 (57.87) | <0.0001 | Ref |
| Does not participate | 9206 (69.28) | 4277 (30.72) | | 0.32(0.29–0.35)** |
| **Partner's highest level of education**ᵐ | | | | |
| No formal education | 6236 (87.43) | 905 (12.57) | <0.0001 | Ref |
| Primary education | 1719 (60.69) | 1178 (39.31) | | 4.50(3.86–5.24)** |
| Secondary education | 3063 (43.03) | 3997 (56.97) | | 9.21(7.90–10.73)** |
| Higher | 888 (28.87) | 2151 (71.13) | | 17.13(14.35–20.45)** |
| **Community poverty** | | | | |
| Low | 3733 (37.34) | 6559 (62.66) | <0.0001 | Ref |
| Medium | 2974 (70.92) | 1325 (29.08) | | 0.24(0.19–0.31)** |
| High | 6084 (85.23) | 1117 (14.77) | | 0.10(0.09–0.14)** |
| **Community education** | | | | |
| Low | 6544 (90.08) | 741 (9.917) | <0.0001 | Ref |
| Medium | 4195 (58.38) | 3058 (41.62) | | 6.48(5.32–7.88)** |
| High | 2052 (26.87) | 5202 (73.13) | | 24.72(20.67–29.56)** |
| **Community distance to health facility** | | | | |
| Low | 3453 (49.5) | 3828 (50.5) | <0.0001 | Ref |
| Medium | 4132 (57.08) | 3187 (42.92) | | 0.74(0.60–0.92)* |
| High | 5206 (71.35) | 1986 (28.65) | | 0.39(0.31–0.49)** |
| **Place of residence** | | | | |
| Urban | 2955 (37.88) | 4755(62.12) | <0.0001 | Ref |
| Rural | 9838 (72.70) | 4246 (27.30) | | 0.23(0.20–0.26)** |
| **Region** | | | | |
| Northcentral | 1878 (49.63) | 1997 (50.37) | <0.0001 | Ref |
| Northeast | 3337 (73.32) | 1169 (26.68) | | 0.36(0.29–0.44)** |
| Northwest | 5320 (83.64) | 989 (16.36) | | 0.19(0.15–0.24)** |
| Southeast | 478 (19.2) | 1887 (80.80) | | 4.15(3.23–5.32)** |
| South-south | 1176 (49.1) | 998 (50.90) | | 1.02(0.83–1.26) |
| South west | 602 (23.8) | 1961 (76.20) | | 3.16(2.57–3.88)** |

ᵐVariables with missing data

*$p<0.05$

**$p<0.001$

Our study is the first study in Nigeria to assess the trends of health facility delivery across the six geopolitical zones. Fasina et. al examined the trends of health facility delivery in Nigeria across two time points—NDHS 2013 and NDHS 2018 for women of reproductive age and found an increase from 38.02% to 42.04% [22]. Our study showed a similar finding of 36.57% in 2008, 37.44% in 2013 and 41.14% in 2018. The slight differences in result could be adduced to differences in the management of variables e.g., handling of variables with missing values. A study from Kenya showed higher prevalence of health facility delivery compared to our study, reporting an increase from 68.3% in 2003 to about 95% in 2015 [23]. In Senegal, facility-based delivery rose from 47% in 1993 to 73% in 2014 and in Namibia, it rose from 67% in 1992 to

**Table 3. Multilevel analysis showing determinants of health facility delivery among women of reproductive age in Nigeria (NDHS 2018).**

| Variables | Model 1 | Model 2 | Model 3 | Model 4 |
|---|---|---|---|---|
| | Empty Model | Individual variables | Community variables | Individual/Community variables |
| | | Adjusted Odds ratio(95% CI) | Adjusted Odds ratio(95% CI) | Adjusted Odds ratio(95% CI) |
| **Age at last childbirth** | | | | |
| 15–19 years | | 1 | | 1 |
| 20–29 years | | 0.90 (0.76–1.07) | | 0.86 (0.72–1.02) |
| 30–39 years | | 1.14 (0.93–1.41) | | 1.07 (0.87–1.32) |
| 40–49 years | | 1.25 (0.95–1.65) | | 1.16 (0.88–1.53) |
| **Religion** | | | | |
| Christianity | | 1 | | 1 |
| Islam | | 0.96 (0.81–1.14) | | 1.06 (0.89–1.27) |
| Traditional/others | | 0.62 (0.34–1.12) | | 0.87 (0.48–1.57) |
| **Ethnicity** | | | | |
| Hausa/Fulani | | 1 | | 1 |
| Yoruba | | 4.15 (3.18–5.40) *** | | 1.21 (0.89–1.65) |
| Ibo | | 9.46 (6.90–12.98) *** | | 3.08 (2.11–4.49) *** |
| Others | | 2.25 (1.87–2.71) *** | | 1.50 (1.24–1.83) *** |
| **Highest level of education** | | | | |
| No formal education | | 1 | | 1 |
| Primary education | | 1.36 (1.17–1.58) *** | | 1.24 (1.07–1.44) ** |
| Secondary education | | 1.88 (1.60–2.20) *** | | 1.67 (1.42–1.95) *** |
| Higher | | 4.40 (3.36–5.76) *** | | 3.82 (2.92–4.99) *** |
| **Employment status** | | | | |
| Unemployed | | 1 | | 1 |
| Employed | | 1.05 (0.93–1.17) | | 1.05 (0.94–1.18) |
| **Wealth index** | | | | |
| Poor | | 1 | | 1 |
| Middle | | 1.71 (1.47–1.99) *** | | 1.32 (1.12–1.54) ** |
| Rich | | 2.90 (2.39–3.50) *** | | 1.90 (1.55–2.33) *** |
| **Exposure to mass media** | | | | |
| Not exposed/infrequent exposure | | 1 | | 1 |
| Frequent exposure | | 1.35 (1.21–1.51) *** | | 1.34 (1.20–1.50) *** |
| **Wanted index pregnancy** | | | | |
| Wanted | | 1.17 (1.01–1.51) * | | 1.15 (0.99–1.34) |
| Not wanted | | 1 | | 1 |
| **Number of childbirths** | | | | |
| 1–2 | | 1 | | 1 |
| 3–4 | | 0.77 (0.67–0.87) *** | | 0.77 (0.68–0.87) *** |
| ≥ 5 | | 0.67 (0.57–0.78) *** | | 0.68 (0.58–0.79) *** |
| **Number of antenatal visits** | | | | |
| < 4 visits | | 1 | | 1 |
| At least 4 visits | | 4.17 (3.63–4.79) *** | | 4.03 (3.51–4.62) *** |
| **Companionship to health facility** | | | | |
| A big problem | | 0.80 (0.69–0.93) ** | | 0.84 (0.72–0.98) * |
| Not a big problem | | 1 | | 1 |
| **Participates in healthcare decision**. | | | | |
| Participates | | 1.12 (1.00–1.25) * | | 1.09 (0.98–1.22) |
| Does not participate | | 1 | | 1 |

*(Continued)*

**Table 3.** (Continued)

| Variables | Model 1 | Model 2 | Model 3 | Model 4 |
|---|---|---|---|---|
| | **Empty Model** | **Individual variables** | **Community variables** | **Individual/Community variables** |
| | | **Adjusted Odds ratio(95% CI)** | **Adjusted Odds ratio(95% CI)** | **Adjusted Odds ratio(95% CI)** |
| **Partner's highest level of education** | | | | |
| No formal education | | 1 | | 1 |
| Primary education | | 1.32 (1.11–1.56) | | 1.21 (1.02–1.44) * |
| Secondary education | | 1.56 (1.33–1.82) | | 1.43 (1.23–1.68) *** |
| Higher | | 2.35 (1.92–2.88) | | 2.14 (1.75–2.62) *** |
| **Community poverty** | | | | |
| Low | | | 1 | 1 |
| Medium | | | 0.52 (0.40–0.67) *** | 0.74 (0.59–0.93) * |
| High | | | 0.33 (0.24–0.44) *** | 0.63 (0.48–0.83) ** |
| **Community education** | | | | |
| Low | | | 1 | 1 |
| Medium | | | 3.73 (2.78–5.01) *** | 1.72 (1.32–2.24) *** |
| High | | | 12.33 (8.38–18.14) *** | 2.65 (1.88–3.72) *** |
| **Community distance to health facility** | | | | |
| Low | | | 1 | 1 |
| Medium | | | 0.80 (0.69–0.97) | 0.88 (0.73–1.05) |
| High | | | 0.52 (0.41–0.67) | 0.71 (0.56–0.88) ** |
| **Place of residence** | | | | |
| Urban | | | 1 | 1 |
| Rural | | | 0.80 (0.65–0.97) * | 1.03 (0.86–1.23) |
| **Region** | | | | |
| Northcentral | | | 1 | 1 |
| Northeast | | | 0.44 (0.33–0.58) *** | 0.54 (0.42–0.71) *** |
| Northwest | | | 0.18 (0.13–0.24) *** | 0.25 (0.19–0.34) *** |
| Southeast | | | 2.81 (2.01–3.91) *** | 1.17 (0.77–1.78) |
| South-south | | | 0.20 (0.15–0.28) *** | 0.20 (0.14–0.27) *** |
| Southwest | | | 1.52 (1.13–2.04) ** | 1.18 (0.86–1.62) |
| **Variance** | 1.212 (0.751–1.957) *** | 0.788 (0.424–1.465) *** | 1.191 (0.742–1.810) *** | 0.757 (0.401–1.431) *** |
| **ICC (%)** | 71.69 | 39.11 | 44.93 | 35.20 |
| **Log Likelihood** | -9723.7591 | -8352.1517 | -8998.7979 | -8174.3727 |
| **Model fit Statistics** | | | | |
| **AIC** | 19453.52 | 16758.30 | 18027.60 | 16426.75 |
| **BIC** | 19477.21 | 16971.46 | 18146.03 | 16734.64 |

*p<0.05
**p<0.01
***p<0.001

87% in 2013 [24]. Despite the positive trend observed in our study, the prevalence of health facility delivery is much lower than these African countries. Even though the efforts to improve maternal healthcare utilization is yielding results in Nigeria, there need for intensified efforts to improve health facility delivery and ultimately delivery outcomes.

From our study, all geopolitical zones showed an increase in health facility delivery prevalence across the years, with exemption of South-south zone where the prevalence of health facility delivery remained relatively the same in 2013 and 2018. Prevalence of health facility

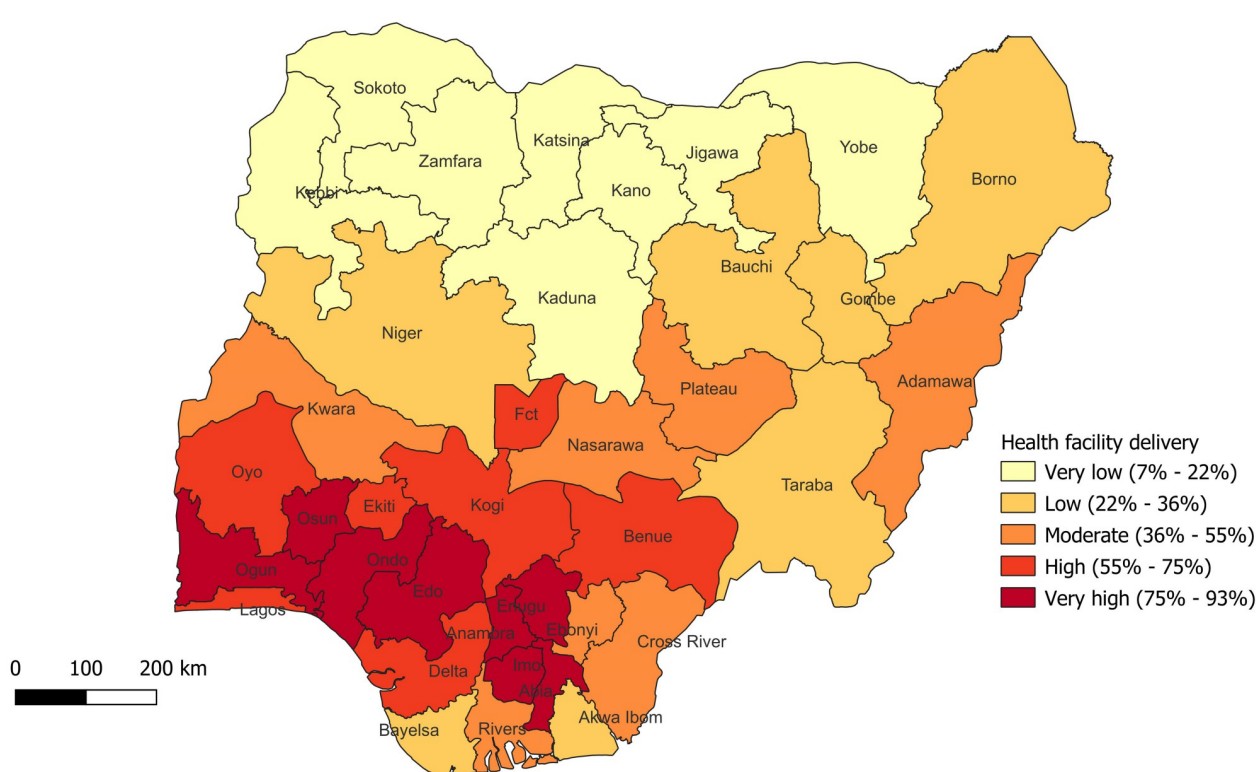

**Fig 2. Percentile map showing the prevalence of health facility delivery among women of reproductive age in Nigeria (NDHS 2018).** Shapefile source: Humanitarian data exchange (HDX); URL: https://data.humdata.org/dataset/cod-ab-nga?

delivery was least in the northern geopolitical zones and the South-south. These may be due to higher levels of poverty, illiteracy, and sociocultural beliefs that may contribute to the low utilization of health facilities for delivery. Strategies to improve health facility delivery in areas with the lowest utilization rates should therefore address poverty and illiteracy, as this study also identified community level poverty and education as determinants of health facility delivery.

On conducting bivariate analysis, we found that all explanatory variables were significantly associated with health facility delivery. Similar findings were reported in other NDHS studies [11, 12]. On further analysis, the study found that as the level of education increased, the women were more likely to choose health facility delivery. This finding is consistent with previous studies, including those that analysed the DHS across the African sub-continent [25–29] and in other LMICs [30, 31]. Education has been shown to promote women's independence in making better health decisions, and highly educated mothers are more likely to seek better health services. Low maternal education has been a significant barrier to health facility delivery and seeking skilled attendants during childbirth [25, 29]. Programmes to improve health literacy among women can improve the use of health facilities for delivery while promoting and ensuring the education of the girl child and women. Another finding of this study is the significant association between wealth index and health facility delivery. As the household wealth index of women increased, the likelihood of facility delivery also increased, this result is in agreement with earlier studies from Nigeria and Ghana [8, 27]. One possible explanation for this finding is that poorer households are prone to financial constraints that may hinder care seeking at health facilities with skilled attendants at delivery. To reduce disparities in access to healthcare, it is critical to reduce financial barriers for pregnant women. This can be achieved

through a functional and efficient health insurance scheme and support for poor women to access health facility during pregnancy.

Access to media also had a significant association with health facility delivery, with the probability of facility delivery increasing with access to media. Pregnant women who have access to media are more likely to obtain valuable information on the benefits of health facility delivery. This finding suggests that the media plays a significant role in providing education and health information, which can considerably influence health-seeking behaviour.

It was found that the probability of health facility delivery reduced as number of childbirths increased. The finding agrees with that of a study from East Africa [32]. The significance of these findings is that women with more children may assume themselves to be more experienced with childbirth, making them choose to deliver at home rather than seek skilled delivery. Another possible reason could be that having a large family size means fewer resources for seeking healthcare, not only for the children within the household but also for the pregnant mother herself. Additionally, if women had a negative experience with health workers during a previous health facility delivery, they may choose to avoid having another health facility delivery. Therefore, it is crucial to provide proper training and supervision for health workers to ensure they demonstrate the right conduct in their provision of maternal care.

Our study revealed that women who had at least four ANC visits were more likely to deliver in the health facility, in keeping with findings from other studies [33, 34]. Optimal ANC attendance provides the opportunity for the women to receive information on available maternal health care services and the benefits of utilizing them, unlike women with no such opportunity. Similarly, our study found that women who reported getting support to go to a health facility was a big problem had a lower probability of delivering in a health facility. Spousal and family support can improve health-seeking for facility delivery and utilization of maternal services health facilities.

Our study also established that women from communities with medium and high community poverty had reduced probabilities of delivering in a health facility, unlike those from communities with low community poverty. Likewise, compared with women from communities with low levels of female education, women from communities with medium levels of female education and those from communities with high levels of female education had higher possibilities of delivering in a health facility. This finding is in consistent with that of a previous study [25]. This finding reveals the need for programmes and interventions that can reduce community poverty and improve communities' socioeconomic status and education level. The study found that women from communities who reported distance to health facilities as a big problem had lower prospects of health facility delivery in keeping with findings of previous studies [8, 25]. A possible explanation for this finding is that when health facilities are not easily accessible, ready alternatives are likely to be preferred to avoid the anticipated difficulties associated with seeking maternal health services from health facilities.

Further still, it was observed in this study that women from the Northeast, Northwest and South-south regions were less likely to utilize health facilities for delivery when compared to women from the Northcentral region of the country. This can also be observed from the spatial analysis in which unlike the north, the southwestern and southeastern states had high and very high utilization of health facility for delivery, and this was seen to decrease moving towards the northern part of the country. These findings agree with that of a previous study [11] and could possibly be due to contextual disparities. These regions have varying levels of poverty, illiteracy, and sociocultural beliefs that may contribute to the findings [35].

Interventions to improve maternal health service utilization in Nigeria include the midwives service scheme and the SURE-P programme which aimed to improve availability of skilled birth attendants in rural areas and provided conditional cash transfer for mothers

utilizing health facilities. These programmes however have not retained the same momentum they had when they started in 2009 and 2012 respectively. There have also been pockets of interventions over time by NGOs, and state governments with the aim of improving maternal health and service utilization [36, 37]. Though some programmes recorded improved outcomes [37–40], others recorded marginal or no improvement in outcomes [41–44]. It is difficult to ascertain if these pockets of interventions affect the utilization of health facility delivery across states and regions as seen in the spatial analysis and regional trends in this study. For example, many interventions were carried out in Northern Nigeria [40, 41, 45–47] which still has lower health facility delivery rates. However, across geo-political zones, health facility delivery has increased over time, from 2008 to 2018, with the exception of South-south zone were rates increased from 2008 to 2013, and then remained same in 2018.

## Implications for policy and practice

As prevalence of health facility delivery was lower in the northern regions which have lower education and higher poverty levels, and community level education and community level poverty were identified determinants of place of delivery in this study, it is essential to intensify efforts to promote formal education and reduce poverty in these settings. Free or subsidized education, advocacy with community leaders on the importance of education, especially girl child education, and poverty alleviation schemes are needed.

This study showed that access to media can significantly influence health-seeking behaviour as women can obtain reliable health information from such channels. Government and non-Governmental agencies should utilize this means more, to pass on health messages. Intensification of family planning promotion campaigns are needed, in order to improve maternal health. This may also affect facility-delivery rates as women with more children are less likely to patronize orthodox care for delivery. Antenatal care attendance should also be promoted. This can be achieved possibly by intensifying community-based activities and mobilization to promote antenatal care utilization, as well as improving health worker attitude and making primary healthcare more accessible to people.

Socio-cultural and religious beliefs and practice may explain why health facility delivery was more among certain ethnic groups e.g. the Igbo ethnic group and lower among others e.g the Hausa/Fulani/Kanuri ethnic group [13]. Socio-cultural beliefs may also to some extent, explain regional differences in health facility delivery. Measures to improve health facility delivery must seek to dispel and eliminate beliefs and value systems that promote home delivery or delivery with unskilled attendants. Future studies should use qualitative methods to further explore factors associated with facility delivery, especially those associated with geopolitical ethnic differences and socio-cultural differences. Future studies are also needed to examine the impact of recent health policies on maternal health.

## Strengths and limitations

This study is the first to examine the trends of health facility delivery across geopolitical zones in Nigeria and is also the first study to conduct a spatial analysis of health facility delivery in the country. However, there are still some limitations to be acknowledged. Firstly, the data was self-reported and collected retrospectively, thus study is susceptible to recall bias. Secondly, the study dataset was cross-sectional in nature, it was only possible to establish an association and not causality. Thirdly, due to multicollinearity, we couldn't determine the effect of marital status on health facility delivery.

## Conclusion

The current study revealed the low prevalence of health facility delivery among women of reproductive age in Nigeria. It further showed the influence of factors such as achieving higher levels of education, having fewer children, having optimal ANC attendance, and many other factors have on health facility delivery. There is a need to deploy appropriate strategies and programme to improve health facility delivery.

## Supporting information

**S1 File. STATA do file.**
(PDF)

## Author Contributions

**Conceptualization:** Tope Olubodun.

**Formal analysis:** Tope Olubodun, Oluyemi Adewole Okunlola.

**Investigation:** Tope Olubodun, Olorunfemi Akinbode Ogundele, Turnwait Otu Michael, Oluyemi Adewole Okunlola, Ayodeji Bamidele Olubodun, Semiu Adebayo Rahman.

**Methodology:** Tope Olubodun, Olorunfemi Akinbode Ogundele, Ayodeji Bamidele Olubodun, Semiu Adebayo Rahman.

**Project administration:** Tope Olubodun.

**Software:** Tope Olubodun.

**Writing – original draft:** Tope Olubodun, Olorunfemi Akinbode Ogundele, Turnwait Otu Michael, Oluyemi Adewole Okunlola, Ayodeji Bamidele Olubodun.

**Writing – review & editing:** Tope Olubodun, Olorunfemi Akinbode Ogundele, Turnwait Otu Michael, Oluyemi Adewole Okunlola, Ayodeji Bamidele Olubodun, Semiu Adebayo Rahman.

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
