## [Decision Letter · Decision Letter 0]

22 May 2024

PONE-D-23-37668Regional Trends, Spatial Patterns and Determinants of Health Facility Delivery Among Women of Reproductive Age in Nigeria: A National Population Based Cross-Sectional StudyPLOS ONE

Dear Dr. Olubodun,

Thank you for submitting your manuscript to PLOS ONE. After careful consideration, we feel that it has merit but does not fully meet PLOS ONE’s publication criteria as it currently stands. Therefore, we invite you to submit a revised version of the manuscript that addresses the points raised during the review process.

We look forward to receiving your revised manuscript.

Kind regards,

Veincent Christian Pepito

Academic Editor

PLOS ONE

Journal Requirements:

Additional Editor Comments:

Dear Authors, thank you very much for your work. The paper is promising and could be published soon. I have a few comments:

1. Clarify if frequencies and percentages in Tables 1 and 2 are weighted or not.

2. Clarify model building strategy - how were variables included in the regression models selected?

3. Data from previous studies in Nigeria are cited, but so should previous studies from other countries, specifically low- and middle-income countries.

https://journals.sagepub.com/doi/full/10.1177/17455057221117957

https://archium.ateneo.edu/asmph-pubs/11/

4. Instead of namedropping previous authors to show the importance of their work, I suggest the authors to just make a blanket declaration: Unlike previous studies,...

5. Show crude odds ratios.

6. When you say multilevel model, do you mean GEE or GLM? Please clarify.

7. The Discussion can be strengthened by putting in more policy recommendations. For example, why are there regional and ethnic disparities in facility birth deliveries? Those who are exposed to mass media are more likely to give birth in facilities. What can government do to address these disparities?

8. Please invert the color of the map. Those who have very low facility births should be dark red while those who have very high facility births should be white because conventionally, those that are red deserve more attention while those that are light colored deserve less.

Reviewers' comments:

Reviewer's Responses to Questions

**Comments to the Author**

1. Is the manuscript technically sound, and do the data support the conclusions?

Reviewer #1: Yes

2. Has the statistical analysis been performed appropriately and rigorously? 

Reviewer #1: I Don't Know

3. Have the authors made all data underlying the findings in their manuscript fully available?

Reviewer #1: No

4. Is the manuscript presented in an intelligible fashion and written in standard English?

Reviewer #1: Yes

5. Review Comments to the Author

Reviewer #1: Q: Has the statistical analysis been performed appropriately and rigorously?

We need to expand on how the analysis (specifically the adjustments for sample weights, clustering, and stratification) was specifically done. Authors may share supplementary material that includes Stata instructions for those interested in reproducing the approach.

Q: Have the authors made all data underlying the findings in their manuscript fully available?

Kindly share all relevant data, unless restrictions on sharing NDHS data are in effect.

Q: Is the manuscript presented in an intelligible fashion and written in standard English?

The manuscript is well written. I do have specific comments and suggestions for some parts. Details will be found in the PDF file I will upload as a "Reviewer Attachment"

6. PLOS authors have the option to publish the peer review history of their article (what does this mean?). If published, this will include your full peer review and any attached files.

Reviewer #1: **Yes: **Theo Prudencio Juhani Z. Capeding, MD MSc

---

## [Author Response · Author response to Decision Letter 0]

6 Sep 2024

There are no comments from reviewers this time

---

## [Editor Report · Decision Letter 1]

30 Sep 2024

Regional Trends, Spatial Patterns and Determinants of Health Facility Delivery Among Women of Reproductive Age in Nigeria: A National Population Based Cross-Sectional Study

PONE-D-23-37668R1

Dear Dr. Olubodun,

We’re pleased to inform you that your manuscript has been judged scientifically suitable for publication and will be formally accepted for publication once it meets all outstanding technical requirements.

Kind regards,

Veincent Christian Pepito

Academic Editor

PLOS ONE
---

## [Editor Report · Acceptance letter]

7 Oct 2024

PONE-D-23-37668R1 

PLOS ONE

Dear Dr. Olubodun, 

I'm pleased to inform you that your manuscript has been deemed suitable for publication in PLOS ONE. Congratulations! Your manuscript is now being handed over to our production team.

Kind regards, 

on behalf of

Mr Veincent Christian Pepito 

Academic Editor

PLOS ONE